# Maternal microchimerism at birth associates with reduced odds of non-malarial fever and respiratory tract infections in Tanzanian children

Gitte L. Petersen[1,2], Paul T. Edlefsen[3], Xiaohong Li[4], Robert Morrison[5],
Edward Kabyemela[6], J. Lee Nelson[7,8], Patrick E. Duffy[5], Michal Fried[5],
Whitney E. Harrington[9,10,11]*

1 Department of Translational Type 1 Diabetes Research, Steno Diabetes Center Copenhagen, Herlev, Denmark, 2 Section of Epidemiology, Department of Public Health, University of Copenhagen, Copenhagen, Denmark, 3 Vaccine and Infectious Disease Division, Fred Hutchinson Cancer Center, Seattle, Washington, United States of America, 4 Division of Public Health Sciences, Fred Hutchinson Cancer Research Center, University of Washington, Seattle, Washington, United States of America, 5 Laboratory of Malaria Immunology and Vaccinology, National Institute of Allergy and Infectious Diseases, National Institutes of Health, Bethesda, Maryland, United States of America, 6 School of Diagnostic Medicine, Muhimbili University of Health and Allied Sciences, Dar es Salaam, Tanzania, 7 Translational Science and Therapeutics, Fred Hutchinson Cancer Research Center, Seattle, Washington, United States of America, 8 Department of Medicine, University of Washington, Seattle, Washington, United States of America, 9 Center for Global Infectious Disease Research, Seattle Children's Research Institute, Seattle, Washington, United States of America, 10 Department of Pediatrics, University of Washington School of Medicine, Seattle, Washington, United States of America, 11 Department of Global Health, University of Washington, Seattle, Washington, United States of America

* whitney.harrington@seattlechildrens.org

## Abstract

The presence of maternal cells in the offspring at birth, a phenomenon known as maternal microchimerism, has been previously associated with decreased odds of malaria and respiratory infections in early childhood suggesting a role in immunological responses to infections. Here, we assess the effect of cord blood maternal microchimerism on symptomatic non-malarial infections in Tanzanian children. We conducted a secondary analysis using a nested birth cohort of 52 children from Muheza, Tanzania, with previously measured cord blood maternal microchimerism and longitudinal records on infections in the first four years of life. The associations between maternal microchimerism and symptomatic lower and upper respiratory tract infections, diarrhea, and non-malarial fever were estimated using generalized estimating equation models. In total, 29% of the 52 children in the study screened positive for cord blood maternal microchimerism. Detected versus non-detected maternal microchimerism was associated with 58% lower odds of non-malarial fever (fully adjusted odds ratio (OR): 0.42 [95% CI: 0.18-0.98]) and 28% lower odds of respiratory tract infection (OR: 0.72 [95% CI: 0.53-0.96]). Lower and upper respiratory tract infections contributed equally to the observed association with any symptomatic respiratory tract infections (ORs respectively: 0.81 [95% CI: 0.50-1.31] and 0.71 [95% CI: 0.50-1.01]). We did not find any association between maternal microchimerism

**Data availability statement:** All relevant data are within the paper and its Supporting Information files.

**Funding:** This study was made possible by a grant from the Novo Nordisk Foundation (NNF18OC0052220 to GLP) and from the National Institute of Allergy and Infectious Diseases (K08 AI135072 to WEH) and Burroughs Wellcome Fund (CAMS 1017213 to WEH). Support for the MMc measurements from the MOMS cohort was provided by the Thrasher Research Fund (to WEH); the National Institutes of Health (T32 HD007233 to WEH); and the Intramural Research Program of the National Institute of Allergy and Infectious Diseases, National Institutes of Health (to MF). Funding for the original MOMS Project was provided by the Bill & Melinda Gates Foundation (29202 to PED), the Foundation for the National Institutes of Health through the Grand Challenges in Global Health Initiative (1364 to PED), the US National Institutes of Health Fogarty International Center (FIC) (D43 TW005509 to PED), and the National Institute of Allergy and Infectious Diseases (R01AI52059 to PED). The funders had no role in study design, data collection and analysis, decision to publish, or preparation of the manuscript.

**Competing interests:** The authors have declared that no competing interests exist.

and odds of diarrhea (OR: 1.63 [95% CI: 0.85-3.13]). Detectable cord blood maternal microchimerism was associated with lower odds of non-malarial fever and symptomatic respiratory infections in Tanzanian infants. These findings emphasize that MMc may play an underrecognized role in protection from infection during early childhood.

## Author summary

During pregnancy, a small number of cells from the mother can cross the placenta and remain in the child's body for many years. This naturally occurring phenomenon is called maternal microchimerism. Previous research has suggested that these maternal cells might influence how a child's immune system responds to infections. In this study, we examined whether the presence of maternal cells in cord blood was linked to the odds of symptomatic non-malarial infections (specifically lower and upper respiratory tract infections, diarrhea, and fever) in young Tanzanian children. Nearly one third of the children tested positive for maternal microchimerism at birth, and these children displayed fewer episodes of fever and symptomatic respiratory infections, compared to children without detectable maternal microchimerism. We did not find any association with odds of diarrhea. Our findings suggest that maternal microchimerism may play an underrecognized role in protection against infection during early childhood.

## Introduction

In 2022, the World Health Organization estimated that the global under-five mortality rate was 37 deaths per 1,000 live births [1]. Sub-Saharan African countries continue to have mortality rates exceeding the United Nations' Sustainable Development Goal of 25 deaths per 1,000 live births by 2030 [2]. Infectious diseases such as lower respiratory tract infections (LRTI), diarrhea, and malaria are leading causes of under-five mortality in these regions [3], and even among survivors, these infections can have long-term consequences for health and functional capacity later in life [4].

Immunological development begins in utero, with immature innate and regulatory cellular responses predominating during early infancy. In the first few years of life, both innate and adaptive responses mature, corresponding to reduced infectious disease burden [5]. During the critical period of early childhood, maternal immunity acquired in utero and via breastmilk in the form of both antibodies and cells, a phenomenon known as maternal microchimerism (MMc), may play a role in protection from infection [6]. During gestation, maternal cells enter the fetal circulation as early as second trimester of pregnancy [7], where they migrate to fetal immune and peripheral organs, and may shape immune development [8]. After birth, MMc is found in a variety of offspring immune cells, including both CD4 and CD8 T cells [9,10]. MMc is proposed to influence offspring immune function through accelerated maturity of innate responses [11] or through the transmission of antigen-specific T cell responses

[12]. Compared with the short half-life of maternal antibodies, MMc may have long-term impact on offspring immunity as it is maintained into adulthood [10,13].

We previously found that detectable cord blood MMc was associated with increased odds of *Plasmodium falciparum* infection but protection from symptomatic disease in young Tanzanian children, suggesting that MMc may modulate risk of infection [14]. Subsequently, Stelzer *et al.* demonstrated an association between detectable cord blood MMc and protection from respiratory infections in German male infants 7–12 months of age, but not younger male infants or female infants, indicating that the impact of MMc may vary by offspring age and sex [11]. In the present study, we extend our analysis of Tanzanian children to determine how cord blood MMc influences odds of symptomatic non-malarial infections non-malarial infections in early childhood, specifically LRTI and upper respiratory tract infections (URTI), diarrhea, and fever. In addition, we assess potential dose-dependent effects, interaction with offspring sex, and effect measure modification by age.

## Materials and methods

### Ethics statement

The establishment of the cohort was approved by both United States (Western Institutional Review Board Study 1059357) and Tanzanian (National Institute for Medical Research, Medical Research Coordinating Committee) ethical review boards. All women provided written informed consent for themselves and their infants to participate.

### Study population

We present a secondary analysis of the Mother Offspring Malaria Study birth cohort conducted in Muheza, Tanzania between September 9th, 2002 and May 15th, 2006 [15]. The study enrolled 882 mother-infant pairs during their delivery hospitalization at the Muheza Designated District Hospital in Tanzania if the mother was 18–45 years of age, free from chronic illness, and had a singleton live birth. Cord blood samples were obtained at delivery and children were subsequently seen at routine visits every two weeks for their first two years of life and every four weeks thereafter up to five years of age. In addition to the routine visits, children were seen at any time of disease symptoms and to ensure parasitemic clearance following a malarial event. All visits involved clinical examination and a blood smear [15,16]. For the present study, we assessed risk of symptomatic non-malarial infections amongst 52 young children with previously measured cord blood MMc levels [14]. All study visits with a positive blood smear (i.e., a *Plasmodium falciparum* parasite count by microscopy greater than zero) were excluded from the main analyses to ensure that the clinical presentation was unrelated to malarial infection. Children were followed from birth up to four years of age, migration, withdrawal of consent, or death, whichever occurred first.

### Maternal microchimerism

Level of cord blood MMc was previously measured [14]. Briefly, MMc was identified by targeting a non-shared, non-inherited maternal polymorphism in genomic DNA (gDNA) extracted from cord blood. When informative human leucocyte antigen (HLA) polymorphisms were not available, four non-HLA loci (ATIII, TG, GSTT1, TNN) targeting insertion/deletion/substitution variants were used to identify a maternal-specific marker. Maternal-specific polymorphisms were then amplified from cord-blood gDNA using qPCR assays with a sensitivity of 1 maternal genomic equivalent (gEq) in 20 000 background infant gEq. Each assay included a polymorphism-specific calibration curve to quantify MMc. Samples were also tested for the nonpolymorphic β-globin gene (HBB), with an HBB calibration curve used to determine total gEq per reaction. Only samples with $\geq 10^4$ expected gEq were included. MMc was modeled both as a binary predictor (no vs. any MMc) and as a normalized continuous predictor (MMc gEq per $10^5$ gEq tested) [14].

## Symptomatic infections

Upon each study visit, symptoms of infectious disease were registered by the study personnel (clinician or village health worker) on exam or by parental report. All LRTI diagnoses were validated on physical exam by a study clinician, whereas URTI and diarrhea were defined by the presence of consistent symptoms reported by the study personnel or the parents. Fever was objectively measured by the study personnel and defined as a temperature >38°C. For the present analysis, we created multiple dummy variables indicating a detected event of 1) any symptomatic infection (LRTI, URTI, diarrhea, or fever), 2) any RTI event (LRTI or URTI), 3) LRTI, 4) URTI, 5) diarrhea, or 6) fever. If children had symptoms of multiple presentations at the same visit, an event was registered for each presentation (e.g., LRTI and diarrhea).

## Confounders

The amount of cord blood gEq assessed varied across individuals based on sample availability, and thus this was included in all models as a covariate [14,17]. Additional potential confounders were selected a priori following the 'disjunctive cause criterion' [18] as any pre-exposure covariate that is known to be associated with cord blood MMc and/or the infections of interest. Predictors of cord blood MMc are poorly described, and we therefore present results from two alternative models with different sets of potential confounders in addition to the unadjusted results. In the first model, we adjust for number of cells screened for MMc and placental malaria at delivery. In the second model, we additionally adjust for maternal age at delivery, birth weight, and months of exclusive breastfeeding.

## Statistical analyses

Characteristics of the study population are provided as frequencies with column percentages (categorical variables) or medians with inter-quartile ranges (continuous variables). A Venn diagram was created using the *ggvenn* package to display the overlap of symptomatic LRTI, URTI, diarrheal infections, and fever at the same study visits. Locally estimated scatterplot smoothing (LOESS) curves with 95% confidence intervals (CIs) were created using the *ggplot2* and the *cowplot* packages to show the probability of symptomatic infections (any, any RTI, LRTI, URTI, diarrhea, and fever, respectively) across age (in weeks).

The association between detected cord blood MMc and symptomatic infections was estimated using generalized estimating equation models (GEEs) by means of the *geepack* package. The data were clustered by individual child to account for potential within-child correlation. A binomial outcome, an independent correlation structure, robust standard errors, and fixed scaling were utilized to ensure that the scale parameter of the binomial distribution remained constant across observations. The parameters of the model estimation were exponentiated to obtain odds ratios (ORs) and 95% CIs.

In the primary analyses, we estimated the odds ratios of symptomatic infection for children tested MMc positive compared to negative and according to categories of detected MMc level (0, >0 to <20, or ≥20 maternal gEqs per 10^5 total gEqs screened) with 0 as the reference group, based on approximately equal group sizes. We tested for statistical interaction between MMc and sex (female vs. male) by rerunning the models with inclusion of a product term.

Potential effect measure modification by offspring age was assessed in sub-analysis. First, we reran the main analyses stratified by age periods 0–1 (greatest potential maternal influence, limited environmental exposure), >1–2 (increasing exposure as child starts exploring independently), and 2+years (increased adaptive immunity and resilience). Second, we estimated the combined effect of MMc and age period by creating a new variable with all combinations of MMc and age periods and including it as the independent variable when rerunning the model. The first model tests if cord blood MMc is associated with different odds of symptomatic infections in each of the three different age periods, whereas the second model provides insight into the OR of any combination of MMc and age period relative to the reference group (MMc negative children aged 0–1 year) enabling comparisons across age.

Finally, we conducted sensitivity analyses to assess the robustness of our findings. First, the data were restricted to routine visits to avoid over-representation of visits on indication of infection (e.g., due to clinician-initiated follow-up or parent observing worsening of symptoms). Second, we introduced a quarantine of 7 days following each registered infection to avoid multiple counts of the same event. Third, we relaxed the inclusion criteria to include study visits with a positive blood smear at least 7 days from parasitemia onset to avoid exclusion of non-malarial infections occurring while the child still had parasitemia [14].

All statistical analyses were conducted using RStudio (version 4.3.2).

## Results

Characteristics of the study population are given in Table 1. Almost one third (29%) of the 52 children in the study population screened positive for MMc in cord blood. Children who screened positive for MMc and those who screened negative were overall comparable with respect to the characteristics presented in Table 1, but children screened positive for MMc had slightly fewer unscheduled study visits than those screened negative.

The 52 children contributed 2,385 aparasitemic study visits of which 1,993 were asymptomatic. Symptomatic infections were registered at 392 of these visits: 348 visits with a single symptomatic infection and 44 visits with two concurrent symptomatic infections, yielding a total of 436 infection events. These events comprised 330 RTI events (96 LRTI and 234 URTI), 49 diarrhea events, and 57 fever events.

The Venn diagram (Fig 1) illustrates the distribution of aparasitemic study visits (n = 2,385) with and without symptomatic non-malarial infections. Symptomatic infections were registered at 392 visits: 348 visits with a single symptomatic infection and 44 with two concurrent symptomatic infections, yielding a total of 436 symptomatic infection events. Symptom categories include LRTI (green), URTI (blue), diarrhea (brown), and fever (red). Overlapping regions represent visits with multiple concurrent symptoms.

LOESS curves with 95% CIs (Fig 2) show the frequency of symptomatic non-malarial infections over time. MMc-negative children (n = 37; 1757 study visits) are represented by solid blue lines, and MMc-positive children (n = 14; 628 study visits) by dashed red lines. The probabilities of episodes of LRTI, URTI, diarrhea, and/or fever increased across the first year of life after which they plateaued, followed by a decrease, although the specific pattern varied by type of symptomatic infection (Fig 2).

Detectable MMc in cord blood was associated with 58% lower odds of fever (fully adjusted OR: 0.42 [95% CI: 0.18-0.98]) and 28% lower odds of RTI (fully adjusted OR: 0.72 [95% CI: 0.53-0.96]) when compared to non-detectable MMc. LRTI and URTI contributed similarly to the MMc-RTI association (ORs respectively: 0.81 [95% CI: 0.50-1.31] and 0.71 [95% CI: 0.50-1.01]). Cord blood MMc was not statistically significantly associated with diarrheal events (OR: 1.63 [95% CI: 0.85-3.13]) nor with the overall odds of any symptomatic infection (RTI, diarrheal, or fever events). Results from the MMc categorical analyses did not lend support to a dose-response relationship between estimated MMc level and infections. Confounder adjustments generally had little influence on the effect estimates (Table 2).

We found no consistent statistical MMc-sex interactions. The MMc-RTI association differed for girls and boys (statistical interaction: p = 0.013), but this was driven by LRTI alone. Due to low statistical power, it was not possible to conduct fully adjusted stratified analyses according to sex with LRTI as the outcome. Partial adjustment for number of cells tested and placental malaria suggested that MMc positive girls had lower odds of LRTI compared to MMc negative girls (OR: 0.30 [95% CI: 0.11-0.76]), while there was no association in boys (OR: 1.18 [95% CI: 0.63-2.22]).

Stratification according to age yielded insecure results due to the low statistical power. The overall reduced odds of RTI and fever with detected MMc appeared to be driven mostly by age 0–1 year and for fever additionally by age 2+ years (Table 3). Results for the remaining outcomes with no overall associations with MMc are presented in Table A in S1

**Table 1. Characteristics of the study population.**

| | Cord blood MMc | | All |
|---|---|---|---|
| | **Negative**<br>**37 (71%)** | **Positive**<br>**15 (29%)** | **52 (100%)** |
| Total number of study visits, median (IQR) | 51 (42-57) | 42 (36-49) | 49 (39-56) |
| Number of routine study visits, median (IQR) | 40 (31 –46) | 38 (30 –42) | 40 (31 –44) |
| Age at end of follow-up (weeks), median (IQR) | 176 (136-184) | 154 (132-174) | 172 (134-181) |
| Year of birth | | | |
| 2002 | 24 (65%) | 7 (47%) | 31 (60%) |
| 2003 | 9 (24%) | 7 (47%) | 16 (31%) |
| 2004 | 3 (8%) | 1 (7%) | 4 (8%) |
| 2005 | 1 (3%) | 0 (0%) | 1 (2%) |
| Maternal age at delivery (years), median (IQR) | 24.0 (22.0-27.0) | 23.0 (20.5-27.5) | 24.0 (21.8-27.0) |
| Use of bed net in home at baseline | | | |
| No bed net | 12 (33%) | 9 (64%) | 21 (42%) |
| Treated bed net | 17 (47%) | 3 (21%) | 20 (40%) |
| Untreated bed net | 7 (19%) | 2 (14%) | 9 (18%) |
| *Missing* | *1* | *1* | *2* |
| Rural residential area | | | |
| No | 16 (43%) | 7 (47%) | 23 (44%) |
| Yes | 21 (57%) | 8 (53%) | 29 (56%) |
| Number of prior pregnancies | | | |
| 0 | 8 (22%) | 6 (40%) | 14 (27%) |
| 1 | 12 (32%) | 2 (13%) | 14 (27%) |
| 2+ | 17 (46%) | 7 (47%) | 24 (46%) |
| Any IPTp* doses received | | | |
| No | 6 (18%) | 1 (7%) | 7 (15%) |
| Yes | 27 (82%) | 13 (93%) | 40 (85%) |
| *Missing* | *4* | *1* | *5* |
| Placental malaria at delivery | | | |
| No | 20 (54%) | 6 (40%) | 26 (50%) |
| Yes | 17 (46%) | 9 (60%) | 26 (50%) |
| Sex | | | |
| Girls | 18 (49%) | 8 (53%) | 26 (50%) |
| Boys | 19 (51%) | 7 (47%) | 26 (50%) |
| Birth weight (kg), median (IQR) | 3.2 (3.0-3.2) | 3.0 (2.6-3.2) | 3.1 (2.7-3.2) |
| *Missing* | *1* | *1* | *2* |
| Birth weight <2.5 kg | | | |
| No | 32 (89%) | 11 (79%) | 43 (86%) |
| Yes | 4 (11%) | 3 (21%) | 7 (14%) |
| *Missing* | *1* | *1* | *2* |
| Born in high malaria transmission season | | | |
| No | 16 (43%) | 7 (47%) | 23 (44%) |
| Yes | 21 (57%) | 8 (53%) | 29 (56%) |
| Breastfeeding | | | |
| Exclusive breastfeeding for three months | 2 (5%) | 3 (20%) | 5 (10%) |
| Exclusive breastfeeding for four months | 35 (95%) | 12 (80%) | 47 (90%) |
| Age at last breastfeeding (months), median (IQR) | 24.8 (22.1-26.5) | 22.9 (21.1-24.6) | 24.1 (21.8-26.3) |

*Intermittent preventive therapy for malaria in pregnancy (IPTp) with sulfadoxine-pyrimethamine

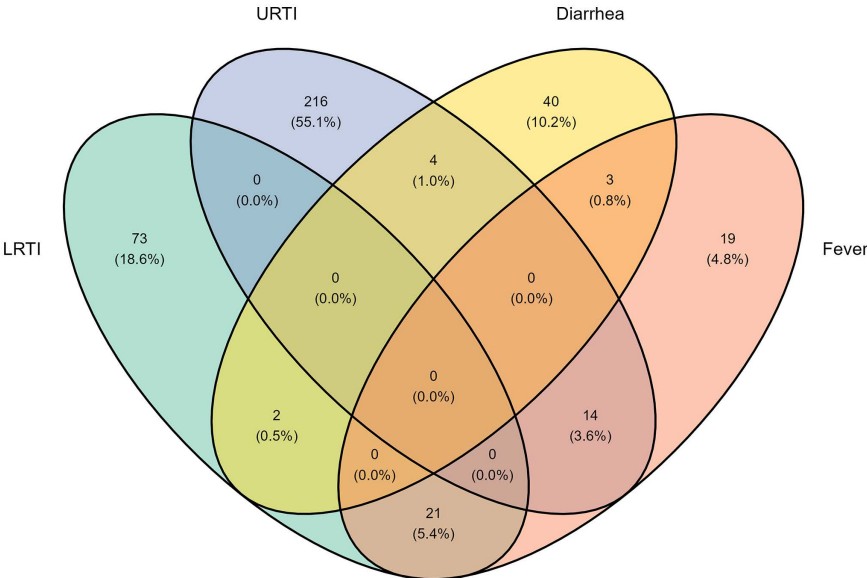

**Fig 1. Venn diagram of symptomatic non-malarial infections among aparasitemic visits.**

Appendix. The joint effect of detected MMc and age (Table B in S1 Appendix) suggested that the odds of RTI in general decreased with age regardless of MMc status.

Overall, the sensitivity analyses supported our conclusions. First, restriction to routine visits (Table C in S1 Appendix) led to slightly attenuated MMc-LRTI association estimates, but with no changes in relation to URTI. The reduced statistical power prevented us from estimating an association between MMc and fever. Second, introduction of a 7-days quarantine for recurrent events (Table D in S1 Appendix) led to exclusion of few events and caused only minor changes to the association measures. Third, inclusion of visits with detected parasitemia (Table E in S1 Appendix) marginally attenuated the association measures for most considered outcomes.

## Discussion

Our findings suggest that Tanzanian children screened positive for cord blood MMc were 28% less likely to experience symptomatic RTI and 58% less likely to experience non-malarial fever in comparison to their counterparts screened MMc negative. We did not detect any association between MMc and diarrhea, and our findings did not lend support to a dose-response relationship between estimated MMc level and odds of symptomatic infections nor to statistical interaction with sex.

To the best of our knowledge, no prior studies have investigated the association between MMc and non-malaria fever during early life. By comparison, our findings with regard to RTI are consistent with the association between cord blood MMc and protection from respiratory infections reported by Stelzer et al. [11], although we identified a potentially stronger effect amongst girls whereas Stelzer et al. found stronger associations in boys. The divergent findings may be due to chance considering the relatively small sample sizes (56 and 52 children in the Stelzer and our populations, respectively), disparate definition of respiratory infections, or population-specific differences between German and Tanzanian children. Our findings with regard to fever and RTI are also consistent with our own prior description of an association between cord blood MMc and lower odds of symptomatic malaria infection [14], suggesting that MMc may protect from

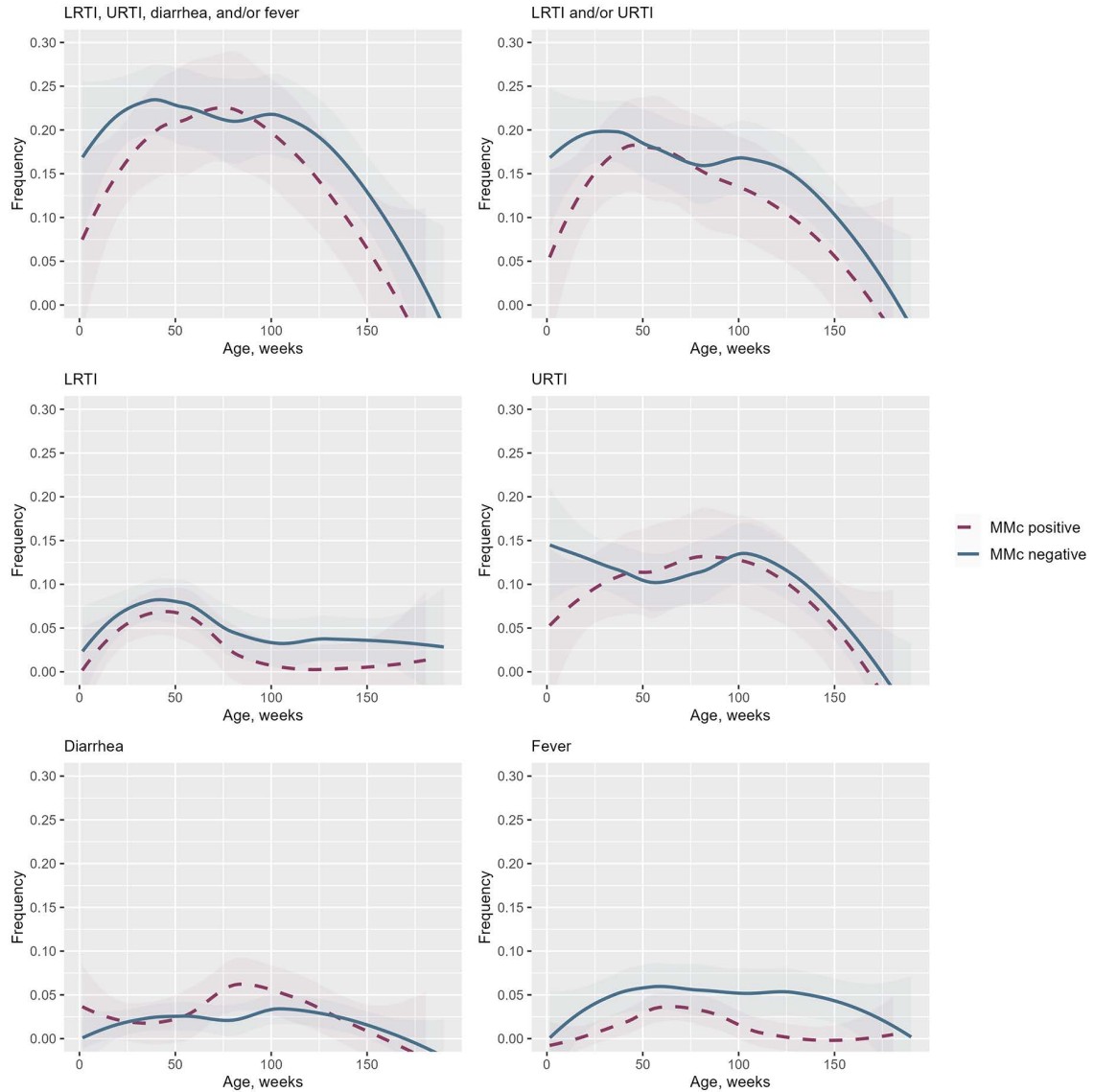

**Fig 2. Symptomatic non-malarial infections by MMc status during the first years of life.**

immunopathology in a non-specific manner. In contrast, we found no association between MMc and diarrheal illness, but our analysis was limited by a low number of events.

Limited work has explored the factors associated with acquisition and retention of MMc in human infants. We have previously found an association between female sex, mother-offspring HLA compatibility, and longer exclusive breastfeeding duration with higher MMc levels in South African infants [12]. In addition, we have found that maternal peripheral [19] and placental malaria [14] are associated with increased cord blood MMc, whereas maternal HIV is associated with decreased MMc [12]. Increased knowledge of population and individual variables associated with MMc will empower more robust understanding of the biological pathways and mechanisms by which MMc may provide infant protection from infection.

Our study had a number of limitations. First, our sample size of 52 children was relatively small and despite the large number of study visits, the analytical numbers were small in some strata. Consequently, our ability to study potential

**Table 2. Odds ratios (ORs) with 95% confidence intervals (CIs) for symptomatic infections according to detected vs not detected maternal microchimerism (MMc) in cord blood or versus estimated quantity of maternal genomic equivalents (gEqs) per 10^5 gEqs tested. The results are from Generalized Estimating Equation models (n = 37 children with no detected MMc*, 14 children with any detected MMc of which 7 had > 0 to <20 and 8 had ≥ 20 MMc gEq/10^5 gEqs tested).**

| MMc test result | Unadjusted | | Adjusted for number of cells tested and placental malaria | | Additionally adjusted for maternal age at delivery, birth weight*, and months of exclusive breastfeeding | |
|---|---|---|---|---|---|---|
| | n visits (n events) | OR (95% CI) | n visits (n events) | OR (95% CI) | n visits (n events) | OR (95% CI) |
| **Any respiratory tract infection, diarrhea, or fever** | | | | | | |
| Not detected | 1757 (300) | 1 (reference) | 1757 (300) | 1 (reference) | 1708 (291) | 1 (reference) |
| Detected | 628 (92) | 0.83 (0.65-1.07) | 628 (92) | 0.82 (0.64-1.05) | 608 (91) | 0.79 (0.60-1.04) |
| Not detected | 1757 (300) | 1 (reference) | 1757 (300) | 1 (reference) | 1708 (291) | 1 (reference) |
| >0 to <20 | 311 (40) | 0.72 (0.50-1.02) | 311 (40) | 0.73 (0.51-1.04) | 311 (40) | 0.71 (0.49-1.02) |
| ≥20 | 317 (52) | 0.95 (0.69-1.32) | 317 (52) | 0.91 (0.66-1.26) | 297 (51) | 0.89 (0.60-1.32) |
| **Respiratory tract infection (upper or lower)** | | | | | | |
| Not detected | 1757 (257) | 1 (reference) | 1757 (257) | 1 (reference) | 1708 (248) | 1 (reference) |
| Detected | 628 (73) | 0.77 (0.58-1.01) | 628 (73) | 0.74 (0.56-0.97) | 608 (73) | 0.72 (0.53-0.96) |
| Not detected | 1757 (257) | 1 (reference) | 1757 (257) | 1 (reference) | 1708 (248) | 1 (reference) |
| >0 to <20 | 311 (31) | 0.65 (0.44-0.96) | 311 (31) | 0.66 (0.44-0.98) | 311 (31) | 0.63 (0.42-0.93) |
| ≥20 | 317 (42) | 0.89 (0.63-1.27) | 317 (42) | 0.82 (0.58-1.17) | 297 (42) | 0.83 (0.54-1.26) |
| **Lower respiratory tract infection** | | | | | | |
| Not detected | 1757 (76) | 1 (reference) | 1757 (76) | 1 (reference) | 1708 (74) | 1 (reference) |
| Detected | 628 20 | 0.73 (0.44-1.20) | 628 [(20) | 0.71 (0.43-1.17) | 608 (20) | 0.81 (0.50-1.31) |
| Not detected | 1757 (76) | 1 (reference) | 1757 (76) | 1 (reference) | 1708 (74) | 1 (reference) |
| >0 to <20 | 311 (9) | 0.66 (0.33-1.33) | 311 (9) | 0.67 (0.33-1.35) | 311 (9) | 0.67 (0.33-1.37) |
| ≥20 | 317 (11) | 0.80 (0.42-1.51) | 317 (11) | 0.76 (0.39-1.45) | 297 (11) | 1.00 (0.52-1.90) |
| **Upper respiratory tract infection** | | | | | | |
| Not detected | 1757 (181) | 1 (reference) | 1757 (181) | 1 (reference) | 1708 (174) | 1 (reference) |
| Detected | 628 (53) | 0.80 (0.58-1.11) | 628 (53) | 0.77 (0.56-1.06) | 608 (53) | 0.71 (0.50-1.01) |
| Not detected | 1757 (181) | 1 (reference) | 1757 (181) | 1 (reference) | 1708 (174) | 1 (reference) |
| >0 to <20 | 311 (22) | 0.66 (0.42-1.05) | 311 (22) | 0.68 (0.43-1.07) | 311 (22) | 0.63 (0.39-1.01) |
| ≥20 | 317 (31) | 0.94 (0.63-1.41) | 317 (31) | 0.86 (0.58-1.29) | 297 (31) | 0.80 (0.48-1.31) |
| **Diarrhea** | | | | | | |
| Not detected | 1757 (31) | 1 (reference) | 1757 (31) | 1 (reference) | 1708 (31) | 1 (reference) |
| Detected | 628 (18) | 1.64 (0.91-2.96) | 628 (18) | 1.73 (0.96-3.10) | 608 (17) | 1.63 (0.85-3.13) |
| Not detected | 1757 (31) | 1 (reference) | 1757 (31) | 1 (reference) | 1708 (31) | 1 (reference) |
| >0 to <20 | 311 (9) | 1.66 (0.78-3.52) | 311 (9) | 1.71 (0.82-3.57) | 311 (9) | 1.95 (0.92-4.12) |
| ≥20 | 317 (9) | 1.63 (0.77-3.45) | 317 (9) | 1.75 (0.83-3.71) | 297 (8) | 1.28 (0.46-3.52) |
| **Fever** | | | | | | |
| Not detected | 1757 (50) | 1 (reference) | 1757 (50) | 1 (reference) | 1708 (50) | 1 (reference) |
| Detected | 628 (7) | 0.38 (0.17-0.85) | 628 (7) | 0.40 (0.18-0.89) | 608 (7) | 0.42 (0.18-0.98) |
| Not detected | 1757 (50) | 1 (reference) | 1757 (50) | 1 (reference) | 1708 (50) | 1 (reference) |
| >0 to <20 | 311 (3) | 0.33 (0.10-1.07) | 311 (3) | 0.32 (0.10-1.04) | 311 (3) | 0.29 (0.08-0.97) |
| ≥20 | 317 (4) | 0.44 (0.16-1.22) | 317 (4) | 0.51 (0.18-1.45) | 297 (4) | 0.68 (0.22-2.09) |

*one child tested MMc negative had missing information on birth weight and was omitted from the fully adjusted analysis

**Table 3. Odds ratios (ORs) with 95% confidence intervals (CIs) for symptomatic respiratory tract infection (RTI) and fever according to detected vs not detected maternal microchimerism (MMc) in cord blood in strata of age. The results are from Generalized Estimating Equation models (n=37 children with no detected MMc*, 14 children with any detected MMc).**

| MMc test result | Unadjusted | | Adjusted for number of cells tested and placental malaria | | Additionally adjusted for maternal age at delivery, birth weight*, and months of exclusive breastfeeding | |
|---|---|---|---|---|---|---|
| | n visits (n events) | OR (95% CI) | n visits (n events) | OR (95% CI) | n visits (n events) | OR (95% CI) |
| **Respiratory tract infection (upper or lower)** | | | | | | |
| Age 0–1 year | | | | | | |
| Not detected | 826 (146) | 1 (reference) | 826 (146) | 1 (reference) | 802 (141) | 1 (reference) |
| Detected | 335 (45) | 0.72 (0.50-1.04) | 335 (45) | 0.68 (0.47-0.98) | 315 (45) | 0.69 (0.47-1.02) |
| Age 1+ to 2 years | | | | | | |
| Not detected | 422 (65) | 1 (reference) | 422 (65) | 1 (reference) | 410 (61) | 1 (reference) |
| Detected | 150 (17) | 0.70 (0.40-1.24) | 150 (17) | 0.69 (0.39-1.22) | 150 (17) | 0.67 (0.38-1.21) |
| Age 2+ years | | | | | | |
| Not detected | 509 (46) | 1 (reference) | 509 (46) | 1 (reference) | 496 (46) | 1 (reference) |
| Detected | 143 (11) | 0.84 (0.42-1.66) | 143 (11) | 0.84 (0.42-1.66) | 143 (11) | 0.94 (0.46-1.94) |
| **Fever** | | | | | | |
| Age 0–1 year | | | | | | |
| Not detected | 826 (22) | 1 (reference) | 826 (22) | 1 (reference) | 802 (22) | 1 (reference) |
| Detected | 335 (2) | 0.22 (0.05-0.94) | 335 (2) | 0.22 (0.05-0.95) | 315 (2) | 0.22 (0.05-1.03) |
| Age 1+ to 2 years | | | | | | |
| Not detected | 422 (12) | 1 (reference) | 422 (12) | 1 (reference) | 410 (12) | 1 (reference) |
| Detected | 150 (5) | 1.18 (0.40-3.49) | 150 (5) | 1.24 (0.44-3.50) | 150 (5) | 1.38 (0.44-4.35) |
| Age 2+ years | | | | | | |
| Not detected | 509 (16) | 1 (reference) | 509 (16) | 1 (reference) | 496 (16) | 1 (reference) |
| Detected | 143 (0) | 0.00 (0.00-0.00) | 143 (0) | 0.00 (0.00-0.00) | 143 (0) | 0.00 (0.00-0.00) |

*One child tested MMc negative had missing information on birth weight and was omitted from the fully adjusted analysis.

interactions was limited. Second, the certainty of our outcome measures may vary across conditions. Fever was measured objectively by trained study personnel, and LRTI diagnoses were validated by a study clinician. In contrast, URTI and diarrhea diagnoses relied more heavily on parental reports, and no international diagnosis classification codes were used in the study. These differences may reduce the certainty of some outcome measures. Differential misclassification is unlikely, as neither parents nor study personnel were aware of infants' MMc status, and we are not aware of any plausible indirect mechanism linking MMc to infection reporting or diagnosis. Replication in cohorts with more standardized infection measures would help confirm our findings. Third, MMc data are known to be skewed with an inflated number of zeros and a few large values [20]. We attempted to address this by estimating associations with both binary and categorial MMc exposure measures [21]. Fourth, in order to determine the association between MMc and non-malarial clinical syndromes, we excluded visits with positive parasitemia. Because MMc was associated with parasitemia in this cohort [14], this resulted in the exclusion of 25% of the available observations from the group of children with detected MMc, whereas this applied to 16% of the observations in those with no detected MMc. Thus, children with frequent malarial infections were less often eligible for consideration with other infections, which may have led to selection bias. Our sensitivity analysis indicated that excluding visits with positive parasitemia might have influenced the estimates. To ensure the robustness of our findings, it would be beneficial to confirm them in additional populations without malaria. Fifth, predictors of cord

blood MMc are not yet well understood, which complicates the identification of confounders such as additional unmeasured comorbidities. Although our alternatively adjusted models yielded largely similar results, unmeasured confounding is possible. Future studies should therefore investigate maternal and infant determinants of cord blood MMc to improve confounder identification and strengthen causal inference. Finally, in the present study we are not able to investigate the mechanism of the association we describe between cord blood MMc and protection from infection, however, this is the subject of ongoing studies.

In conclusion, we describe an association between cord blood MMc and protection from symptomatic respiratory tract infections and non-malaria fever in Tanzanian children. Together, these findings emphasize that MMc may play an under-recognized role in protection from infection during infancy and early childhood.

## Supporting information

**S1 Appendix. Tables A-E.** Table A the estimated associations between maternal microchimerism (MMc) and symptomatic infections in strata of age. Odds ratios (ORs) with 95% confidence intervals (CIs) are from Generalized Estimating Equation models (n=37 children with no detected MMc*, 14 children with any detected MMc). Table B The estimated joint effect of maternal microchimerism (MMc) and age period on probability of symptomatic infections. Odds ratios (ORs) with 95% confidence intervals (CIs) are from Generalized Estimating Equation models (n=37 children with no detected MMc*, 14 children with any detected MMc). Table C Odds ratios (ORs) with 95% confidence intervals (CIs) for symptomatic infections according to detected maternal microchimerism (MMc) in cord blood. The results are from Generalized Estimating Equation models (n=37 children with no detected MMc*, 14 children with any detected MMc). The analyses are restricted to routine study visits. Table D Odds ratios (ORs) with 95% confidence intervals (CIs) for symptomatic infections according to detected maternal microchimerism (MMc) in cord blood. The results are from Generalized Estimating Equation models (n=37 children with no detected MMc*, 14 children with any detected MMc). Recurrent infections within 7 days were excluded. Table E Odds ratios (ORs) with 95% confidence intervals (CIs) for symptomatic infections according to detected maternal microchimerism (MMc) in cord blood. The results are from Generalized Estimating Equation models (n=37 children with no detected MMc*, 14 children with any detected MMc). Infections within 7 days of a malaria diagnosis were excluded if the child was positive for parasitemia. (DOCX)

**S1 Data. Primary data.**
(CSV)

## Acknowledgments

We thank the women and children from Muheza, Tanzania, for their participation in the MOMS cohort and recognize Dr T. K. Mutabingwa and the MOMS clinical team for their care of study participants.

## Author contributions

**Conceptualization:** Gitte L. Petersen, Michal Fried, Whitney E Harrington.

**Data curation:** Gitte L. Petersen, Robert Morrison, Patrick E. Duffy, Michal Fried, Whitney E Harrington.

**Formal analysis:** Gitte L. Petersen, Paul T. Edlefsen, Xiaohong Li, Michal Fried, Whitney E Harrington.

**Funding acquisition:** J. Lee Nelson, Patrick E. Duffy, Michal Fried, Whitney E Harrington.

**Investigation:** Robert Morrison, Edward Kabyemela, Patrick E. Duffy, Michal Fried, Whitney E Harrington.

**Methodology:** Gitte L. Petersen, Paul T. Edlefsen, Xiaohong Li, J. Lee Nelson, Whitney E Harrington.

**Project administration:** Robert Morrison, Edward Kabyemela, Patrick E. Duffy, Michal Fried, Whitney E Harrington.

**Resources:** Edward Kabyemela, J. Lee Nelson, Patrick E. Duffy, Michal Fried, Whitney E Harrington.

**Supervision:** Edward Kabyemela, Patrick E. Duffy, Michal Fried, Whitney E Harrington.

**Writing – original draft:** Gitte L. Petersen, Whitney E Harrington.

**Writing – review & editing:** Gitte L. Petersen, Paul T. Edlefsen, Xiaohong Li, Robert Morrison, Edward Kabyemela, J. Lee Nelson, Patrick E. Duffy, Michal Fried, Whitney E Harrington.

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
