## [Decision Letter · Decision Letter 0]

20 Jan 2026

PGPH-D-25-02597

Maternal microchimerism at birth associates with reduced odds of non-malarial fever and respiratory tract infections in Tanzanian children

Dear Dr. Harrington,

Thank you for submitting your manuscript to PLOS Global Public Health. After careful consideration, we feel that it has merit but does not fully meet PLOS Global Public Health’s publication criteria as it currently stands. Therefore, we invite you to submit a revised version of the manuscript that addresses the points raised during the review process.

Please note that we have only been able to secure a single reviewer to assess your manuscript. We are issuing a decision on your manuscript at this point to prevent further delays in the evaluation of your manuscript. Please be aware that the editor who handles your revised manuscript might find it necessary to invite additional reviewers to assess this work once the revised manuscript is submitted. However, we will aim to proceed on the basis of this single review if possible.

Could you please revise the manuscript to carefully address the concerns raised?

We look forward to receiving your revised manuscript.

Kind regards,

Helen Howard

Staff Editor

Journal Requirements:

Additional Editor Comments (if provided):

Reviewers' comments:

Reviewer's Responses to Questions

**Comments to the Author**

1. Does this manuscript meet PLOS Global Public Health’s publication criteria? Is the manuscript technically sound, and do the data support the conclusions? The manuscript must describe methodologically and ethically rigorous research with conclusions that are appropriately drawn based on the data presented.

Reviewer #1: Yes

2. Has the statistical analysis been performed appropriately and rigorously?

Reviewer #1: Yes

3. Have the authors made all data underlying the findings in their manuscript fully available (please refer to the Data Availability Statement at the start of the manuscript PDF file)?

Reviewer #1: Yes

4. Is the manuscript presented in an intelligible fashion and written in standard English?

Reviewer #1: Yes

5. Review Comments to the Author

Reviewer #1: Petersen et al. present a secondary analysis of a birth cohort comprising 52 children from Tanzania, describing an association between cord blood microchimerism and protection against respiratory tract infections and non-malaria febrile illness during the first four years of life. The authors are to be commended for leveraging a uniquely rich longitudinal dataset, with follow-up conducted every two weeks during the first two years of life and every four weeks thereafter, up to five years. The study is well conducted, the analyses are appropriate, and the manuscript is well written. However, some points require clarification or minor revision to improve transparency, interpretability, and completeness of the manuscript.

Minor revisions

Methods

- Please include the approval number.

- The manuscript states that the study was conducted between 2002 and 2006. Please specify the exact start and end dates (day and month), including recruitment and follow-up details.

- The methods used to assess maternal microchimerism should be described in greater detail in the current manuscript, including laboratory techniques, thresholds for positivity, quality control procedures, and any relevant validation references.

- Please expand the description of how symptomatic infection outcomes were defined. Specifically: (1) who made the diagnoses (study personnel vs. routine healthcare providers), (2) if there were ICD-10 or other standardized diagnostic codes used, and (3) if there were any diagnostic tests performed beyond malaria blood smears. In addition to including this information in the methods section, please comment on it in the discussion and address it in the limitations, as appropriate.

- Additionally, as all infection outcomes refer exclusively to symptomatic infections, please ensure this is stated explicitly and consistently throughout the manuscript, including the abstract.

Results:

- Please clearly report the number of mother–infant pairs included in the original Mother Offspring Malaria Study cohort and the number included in this secondary analysis. If not all participants were analysed, please explain the reasons for non-inclusion. I also suggest inclusion of a participant flow diagram to improve clarity.

- Table 1. Please report the prevalence of relevant maternal and child comorbidities (e.g., HIV infection, tuberculosis, malnutrition), as these may act as important confounders. If such data are unavailable, this limitation should be explicitly acknowledged and discussed.

- In line 182, the manuscript reports 330 respiratory tract infection events, including 96 lower RTIs and 234 upper RTIs, as well as 49 diarrhoea and 57 fever events. However, Figure 1 refers to 392 symptomatic infections in total. This discrepancy is likely due to overlapping events, but this should be clearly explained in the text to avoid confusion.

Figures:

- The current resolution and visual quality of both figures are insufficient. Please improve figure quality and consider using lighter or more contrasting colours in Figure 1 to enhance readability, particularly for text within the figure.

6. PLOS authors have the option to publish the peer review history of their article (what does this mean?). If published, this will include your full peer review and any attached files.

**Do you want your identity to be public for this peer review?** For information about this choice, including consent withdrawal, please see our Privacy Policy.

Reviewer #1: No

Figure Resubmissions:

---

## [Decision Letter · Decision Letter 1]

23 Apr 2026

Maternal microchimerism at birth associates with reduced odds of non-malarial fever and respiratory tract infection in Tanzanian children

PGPH-D-25-02597R1

Dear Dr. Harrington,

We are pleased to inform you that your manuscript 'Maternal microchimerism at birth associates with reduced odds of non-malarial fever and respiratory tract infection in Tanzanian children' has been provisionally accepted for publication in PLOS Global Public Health.

Before your manuscript can be formally accepted you will need to complete some formatting changes, which you will receive in a follow up email. A member of our team will be in touch with a set of requests. Please also see Reviewer 2's suggestion to revise the title of Table 3.

Best regards,

Katia Bruxvoort, PhD

Academic Editor

Please update the title for Table 3, per the reviewer's suggestion.

Reviewer Comments (if any, and for reference):

Reviewer's Responses to Questions

**Comments to the Author**

1. If the authors have adequately addressed your comments raised in a previous round of review and you feel that this manuscript is now acceptable for publication, you may indicate that here to bypass the “Comments to the Author” section, enter your conflict of interest statement in the “Confidential to Editor” section, and submit your "Accept" recommendation.

Reviewer #1: All comments have been addressed

Reviewer #2: All comments have been addressed

2. Does this manuscript meet PLOS Global Public Health’s publication criteria? Is the manuscript technically sound, and do the data support the conclusions? The manuscript must describe methodologically and ethically rigorous research with conclusions that are appropriately drawn based on the data presented.

Reviewer #1: Yes

Reviewer #2: Yes

3. Has the statistical analysis been performed appropriately and rigorously?

Reviewer #1: Yes

Reviewer #2: Yes

4. Have the authors made all data underlying the findings in their manuscript fully available (please refer to the Data Availability Statement at the start of the manuscript PDF file)?

Reviewer #1: Yes

Reviewer #2: Yes

5. Is the manuscript presented in an intelligible fashion and written in standard English?

Reviewer #1: Yes

Reviewer #2: Yes

6. Review Comments to the Author

Reviewer #1: (No Response)

Reviewer #2: This is an important and well-written study. The introduction is succinct and well-structure, allowing the reader to get familiar with the topic and the study objective. The methods section clearly defines the main outcome and exposure variables, and it indicates how they were measured. It discusses how confounders were selected and provides sound rationale for their statistical analyses, including using GEE to analyze longitudinal data linked by participant ID. I appreciate that they include multiple regression models in their results section and they discuss the differences between them. The authors clearly list the limitations of the study and mention the strategies they used to address the when possible.

Table 3: the title should be updated as it presents the association of MMc and both respiratory infections and fever, not just with fever.

7. PLOS authors have the option to publish the peer review history of their article (what does this mean?). If published, this will include your full peer review and any attached files.

**Do you want your identity to be public for this peer review?** For information about this choice, including consent withdrawal, please see our Privacy Policy.

Reviewer #1: No

Reviewer #2: No
